# Variability of tissue mechanical response in *Sus Domesticus* porcine models from *in vivo* to *ex vivo* conditions

**Faizan A. Malik**[1], **Bradley A. Drahos**[1], **Amer M. Safdari**[2], **Mark V. Mazzeo**[3], **Jack E. Norfleet**[3], **Robert M. Sweet**[4], **Timothy M. Kowalewski**[1]*

**1** University of Minnesota Mechanical Engineering, Minneapolis, Minnesota, United States of America,
**2** University of Minnesota Biomedical Engineering, Minneapolis, Minnesota, United States of America, **3** U.S. Army Combat Capabilities Development Command – Soldier Center, Natick, Massachusetts, United States of America, **4** Department of Surgery, University of Washington Division of Healthcare Simulation Science, Seattle, Washington, United States of America

* timk@umn.edu

## Abstract

**Data Availability Statement:** All data used as part of this research can be found in the Data

### Background

Healthcare simulators have been demonstrated to be a valuable resource for training several technical and nontechnical skills. A gap in the fidelity of tissues has been acknowledged as a barrier to application for current simulators; especially for interventional procedures. Inaccurate or unrealistic mechanical response of a simulated tissue to a given surgical tool motion may result in negative training transfer and/or prevents the "suspension of disbelief" necessary for a trainee to engage in the activity. Thus, where it is relevant to training outcomes, there should be an effort to create healthcare simulators with simulated tissue mechanical responses that match or represent those of biological tissues. Historically, this data is most often gathered from preserved (*post mortem*) tissue; however, there is a concern that the mechanical properties of preserved tissue, that lacks blood flow, may lack adequate accuracy to provide the necessary training efficacy of simulators.

### Methods and findings

This work explores the effect of the "state" of the tissue testing status on liver and peritoneal tissue by using a customized handheld grasper to measure the mechanical responses of representative porcine (*Sus domesticus*) tissues in $n = 5$ animals across five test conditions: *in vivo*, *post mortem* (*in-situ*), *ex vivo* (immediately removed from fresh porcine cadaver), post-refrigeration, and post-freeze-thaw cycle spanning up to 72 hours after death. No statistically significant difference was observed in the mechanical responses due to grasping between *in vivo* and post-freeze conditions for porcine liver and peritoneum tissue samples ($p = 0.05$ for derived stiffness at grasping force values F = 5N and 6.5N). Furthermore, variance between *in vivo* and post-freeze conditions *within* each animal, was comparable to the variance of the *in vivo* condition *between* animals.

Repository for University of Minnesota (DRUM) using the following link: https://conservancy.umn.edu/handle/11299/227135.

**Funding:** This work is based on research funded by the DEVCOM Army Research Laboratory (ARL) through the cooperative agreement manager Jack Norfleet under grant W911NF-14-2-0035. https://www.arl.army.mil/. Authors receiving salary or other funding from this grant include FAM, BAD, AMS, and TMK. The funders, outside of those listed as authors from DEVCOM ARL, had no role in study design, data collection and analysis, decision to publish, or preparation of the manuscript.

**Competing interests:** The authors have declared that no competing interests exist.

## Conclusions

Results of this study further validate the use of preserved tissue in the design of medical simulators via observing tissue mechanical responses of post-freeze tissue comparable to *in vivo* tissue. Therefore, the use of thawed preserved tissue for the further study and emulation of mechanical perturbation of the liver and peritoneum can be considered. Further work in this area should investigate these trends further, particularly in regard to other tissues and the potential effects varying preservation methods may yield.

## Introduction

The lack of accurate tissue mechanical behavior is a well-recognized barrier to the wide-spread adoption of simulated analogues for moderate to complex interventional healthcare procedures. The state-of-the-art approach for improving the mechanical characteristics of medical simulators is to model them after data derived from existing literature on tissue biomechanics. A potential limitation is that much of the existing tissue characterization data is measured outside of the tissue's natural environment, often-times long after the host has died. One concern is that by using this pre-existing data based on *ex vivo* tissue, physical and virtual analogues will simply behave more cadaveric rather than mimicking the mechanical properties and behaviors of perfused tissue. Little is currently known about how contemporary biomechanical characterization techniques truly match the properties of the tissues *in vivo*. If this discrepancy was further studied and characterized, it could provide evidence either in support of using existing, widespread *ex vivo* data as suitable for the development of simulated healthcare models or contrary evidence, emphasizing the need for *in vivo* data.

### Prior work

One of the earliest and most notable direct comparisons between *in vivo* and *ex vivo* tissue was conducted by Rosen et al. in 2008 [1]. The purpose of this study was to characterize the changes in tissue mechanical properties measured *ex vivo* to their *in vivo* state to aid in the development of more mechanically realistic medical simulators; that is, a simulated tissue's force-motion profile should resemble that of its targeted living tissue. During this study, Rosen et al. developed a handheld grasping device which could apply compressive loads to tissue and monitor applied stresses and strains. This device was designed such that it could gather data *in vivo* which could then be compared to results from testing tissue *ex vivo* using a uniaxial compressive fixture (MTS). Both sets of data were fit with a phenomenological modeling approach and the key finding was that properties of the tissue had a noticeable difference from *in vivo* to *ex vivo* for most organs.

A similar study, conducted by Mazza et al., analyzed mechanical properties of the human cervix *in vivo* and *ex vivo* both before and after a hysterectomy procedure [2]. For this study a stiffness parameter was used to characterize mechanical properties of the tissue. Around the same time as the previous studies, Ocal et al. verified a common assumption that the longer tissue is preserved *ex vivo*, the stiffer and more viscous it becomes through testing of bovine liver tissue [3]. Additionally, Kerdok et al. analyzed the effects of perfusion on mechanical properties of porcine liver [4]. For this study, a primary source of concern was whether tissue damage would affect tissue mechanical response.

Brouwer et al. in 2001 analyzed mechanical property characterization between *in vivo* and *ex vivo* porcine tissue to provide more accurate haptics for surgical simulation [5]. This study made use of an indentation device that could be used to compress tissues that were too fragile for tensile tests. The resulting data was fit to exponential curves with high fit.

Prior work in this project published by Safdari in 2019 [6] investigated characterizing tissue mechanical response in different states. This study only analyzed the tissues in the *post mortem*, to *ex vivo*, to post-freeze tissue states. The study introduced techniques to analyze the force-strain data (mechanical response) of porcine tissue under multiple conditions; this work aims to extend and optimize those techniques, as well as incorporate the critical *in vivo* tissue state into the analysis. The study, like this work, used a manual, human-operated grasper device. While this does limit repeatability due to inherent human variances in strain rates and force, it does ensure that the tissues are tested within realistic human viscoelastic regimes.

### Research objective and contributions of this paper

This study hopes to combine elements of the aforementioned studies and produce a comprehensive data set by attempting to measure the change in mechanical responses encountered in typical surgical grasping in porcine models between tissue in its native living (*in vivo*) state and four deceased states: *post mortem*, *ex vivo*, post-refrigeration, and post-freeze. Descriptions of the experimental conditions for all tissue states appear in Table 1. The target tissues were limited to intact whole organs or complexes such as liver, spleen, peritoneal tissue (peritoneum), and lung. For each of these tissues, a manual grasping motion following a repeated, audio cued timing cadence was applied to the tissue, and the resulting force-displacement recordings were used to infer and analyze the tissue's stiffness at set compressive loads of F = 5N and 6.5N, that fell in a common range observed during testing. Measured changes in inferred tissue stiffness across all five states (from *in vivo* to thawed) within a given animal were compared to measurements between all $n = 5$ animals. Table 2 compares and summarizes the scope of cited prior works, as well as the scope of the presented study.

## Materials and methods

### Approval and ethics statement

This study was carried out in accordance with the recommendations in the Guide for the Care and Use of Laboratory Animals of the National Institutes of Health. The protocol was approved by the Institutional Animal Care and Use Committee of the University of Minnesota (Protocol Number: 1805-35922A); as well as the USAMRMC Animal Care and Use Review Office (ACURO) under Award Number W911NF-14-2-0035. All surgery was performed under anesthesia, and all efforts were made to minimize suffering.

**Table 1. Tissue testing conditions.**

| Condition | Tissue State | Time After Death [hr.] | in/ex situ |
|:---:|:---:|:---:|:---:|
| 1 | *in vivo* | N/A | in-situ |
| 2 | *post mortem* | 1–2 | in-situ |
| 3 | *ex vivo* | 5–6 | ex-situ |
| 4 | post-refrigeration | 24–36 | ex-situ |
| 5 | post-freeze | 48–72 | ex-situ |

**Table 2. Comparison of prior work.**

| Reference | *in vivo* | *in-situ* | *ex vivo* | Modeling | Test Specimen | Test Type |
|---|---|---|---|---|---|---|
| Rosen et al. [1] | x | | x | | Porcine organs | Tension, indentation |
| Mazza et al. [2] | x | | x | | Human cervix | aspiration |
| Ocal [3] | | | x | maxwell-voigt | Porcine liver | Impact, stress relaxation |
| Kerdok et al. [4] | x | | x | maxwell-voigt | Porcine liver | indentation |
| Brouwer et al. [5] | x | | x | | Porcine organs | Tension, indentation |
| Safdari et al. [6] | | x | x | mooney-rivlin/ogden finite element and model free | Porcine organs | Indentation with hand-held grasper device |
| Current Study | x | x | x | | Porcine organs | Indentation with hand-held grasper device |

## Porcine models and tissue handling

The population for this study consisted of five Yorkshire-X porcine animals. The population size of five was chosen to ensure that inter-animal variability could be represented in our data set, while minimizing the number of animals needed. Weights of each animal ranged from 71.2kg to 83.0kg while the animals ranged between 119 days to 154 days (4 to 5 months) in age. All animals were determined to have no health concerns, as determined by Research Animal Resources laboratory staff of the University of Minnesota. An overview of the study population is shown in Table 3 below.

**Anesthesia.** The animal was given an intramuscular (IM) dose of 5–7 mg/kg Tiletamine/ Zolazepam, an ear vein catheter was started, and the swine was given an additional 5–7 mg/kg methohexital intravenously (IV) and given 0.9% saline through the ear vein during the procedure. The animal was intubated and maintained on isoflurane >1.2 MAC for the remainder of the procedure. An arterial line was placed in a branch of the femoral artery to monitor blood pressure. Throughout testing, Visible Heart Lab (VHL) professionals with anesthesia training were present and monitored for signs of shock or inadvertent emergence from anesthesia, although no such events occurred. The animal was monitored by heart rate, blood pressure, jaw tone, pupil dilation, and movement.

A medial sternotomy was performed using electrocautery and a sternal saw. The sternum was retracted and the pericardium was removed or sutured in the corners to produce a pericardial cradle. Lipovenous or other combinations of omega-3 fatty acids were administered to all animals as necessary to promote viability *in vitro* in all subjects as part of standard protocol in the VHL.

**Euthanasia.** After data collection was complete, the heart was isolated for separate, unrelated *in vitro* research in order to maximize the use of the animal. The swine remained under a deep plane of anesthesia during all procedures prior to euthanasia.

**Table 3. Study population overview.**

| Condition | Gender | Session Start Date | Time of Death | Body Temp (°C) | Weight (kg) | Date of Birth | Age (days) |
|---|---|---|---|---|---|---|---|
| 1 | F | 2019/07/15 | 12:40 PM | 35.6 | 76.4 | 2019/03/07 | 131 |
| 2 | F | 2019/07/24 | 12:40 PM | 36.9 | 80.4 | 2019/03/07 | 140 |
| 3 | F | 2019/08/07 | 9:29 AM | 38.6 | 71.2 | 2019/04/03 | 127 |
| 4 | F | 2019/09/18 | 11:59 AM | 38.1 | 79.2 | 2019/05/23 | 119 |
| 5 | F | 2019/10/23 | 11:40 AM | 38.7 | 83.0 | 2019/05/23 | 154 |

Only Yorkshire-X type pigs were used.

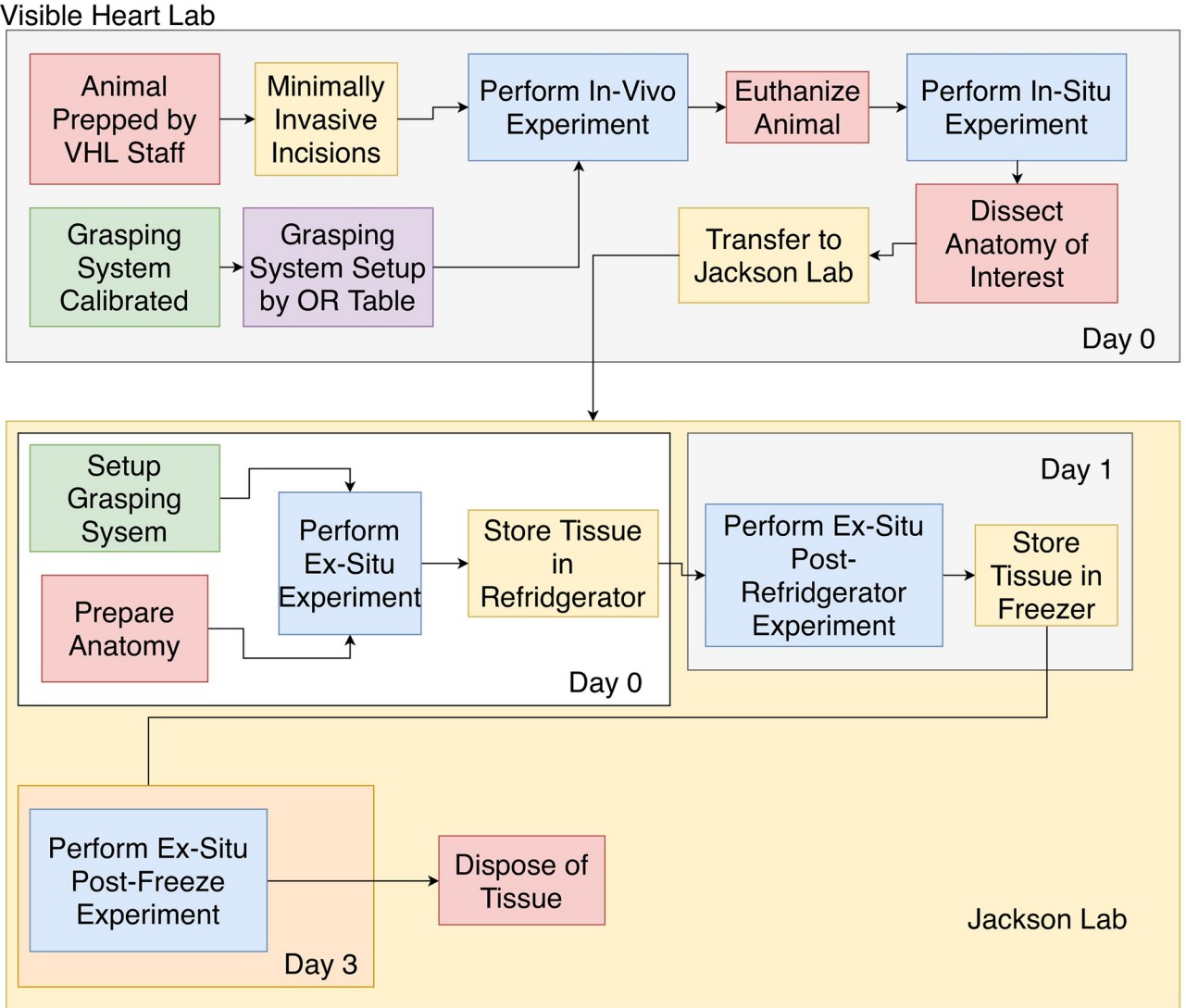

**Fig 1. Tissue handling and testing overview.**

~30,000 units of heparin were administered via IV.; an aortic root cannula was sutured into the ascending aorta; the inferior vena cava and ascending aorta were clamped and high potassium cardioplegia was administered via the aortic root cannula under 150 mmHg pressure. The superior vena cava was clamped, and a small incision was made in the pulmonary artery. The heart was arrested before being excised for other unrelated studies. Any additional procedures, as part of other studies, were planned accordingly by the VHL to minimize risk of cross study and procedure effects.

**Facilities.** The animal experiments were conducted at a dedicated operating facility (Visible Heart Lab) where each animal was used for additional unrelated scientific studies regarding the heart. Immediately after the *post mortem* studies, the tissues were taken to a separate nearby facility (Wet Lab). The procedure is shown in Fig 1.

***In vivo* and *Post mortem*.** An incision was made starting in the abdominal cavity (to access the liver, spleen, and peritoneum tissues), and then later expanded to the chest cavity

(to access the lung and aorta). The body temperature of the animal was measured via a rectal thermometer and recorded.

After the data collection process on the desired tissues was complete, the animal was euthanized, and its heart was extracted using the procedure above. Immediately after euthanasia (time was noted), data collection continued on the same tissues that were tested during *in vivo* conditions.

***Ex vivo*, refrigeration and freezing.** After *post mortem* testing, the tissues were extracted from the carcass and placed in a plastic bag with 25 mL of 1x PBS/pen-strep antibiotic solution. Due to its larger size, the liver was placed in a bag with 75 mL of the same solution.

The plastic bags containing the tissues were held in water baths at 37.5 degrees Celsius until data was collected for the *ex vivo* stage. After *ex vivo* data collection, the tissues were replaced in their respective plastic bags (still containing the antibiotic solution), and refrigerated at approximately 4 degrees Celsius.

After approximately 24 hours of refrigeration, the bags containing the tissues were removed from the refrigerator and reheated in a water bath at 37.5 to 38 degrees C for 2–4 hours. The tissues remained in the antibiotic solution until they were temporarily removed for data collection. After data collection was completed, the tissues were replaced in plastic bags. As the bags became saturated with fluids from the tissue, the bags were replaced as was the antibiotic solution.

The tissues were then placed in a freezer at -18 degrees C for approximately 48 hours, after which they were reheated in a water bath at 38 degrees C for 4–5 hours. Data was collected from the tissues for a final time, after which the tissues were disposed.

The tissue testing conditions are summarized in Table 1.

Actual temperatures were measured at all stages of the protocol stated above; measured temperatures for the liver and peritoneum are shown in S1 Fig.

## Grasping device design

The device used for all data collection was a scissor-like tissue grasping device designed by the Medical Robotics and Devices lab. A detailed discussion of the grasping device and its measurement performance is described in S2 Fig. and Drahos et al. [7]. The device contains two opposing load cells (TAL220B, HT Sensor Technology) in a scissor-like fashion, with 3D-printed hemispherical plastic tips serving as the grasping surface and a quadrature rotary encoder (AMT102-V, CUI Inc.) to measure the angle of the arms as the device opens to monitor tissue thickness and overall displacement. A depiction of the grasper device is shown in Fig 2.

The computer running the Scissors Console application utilized NI DAQMX drivers to communicate with the DAQ and acquire the voltage and encoder data at a frequency of 1 kHz. The Scissors Console application automatically stored the data in CSV text files of the user's choosing, and also displayed a live graph of the data. Fig 3 shows how the data is collected from the sensors.

## Grasping device operation

**Setup.** The grasping device was placed into a medical glove such that the load cells are placed into the fingers of the glove: the top jaw went into the glove's $2^{nd}$ digit ("index finger") and the lower jaw was placed in the glove's $5^{th}$ digit ("pinkie finger"). The remaining fingers of the glove were taped against the device in order to prevent interference during testing, as seen in Fig 4.

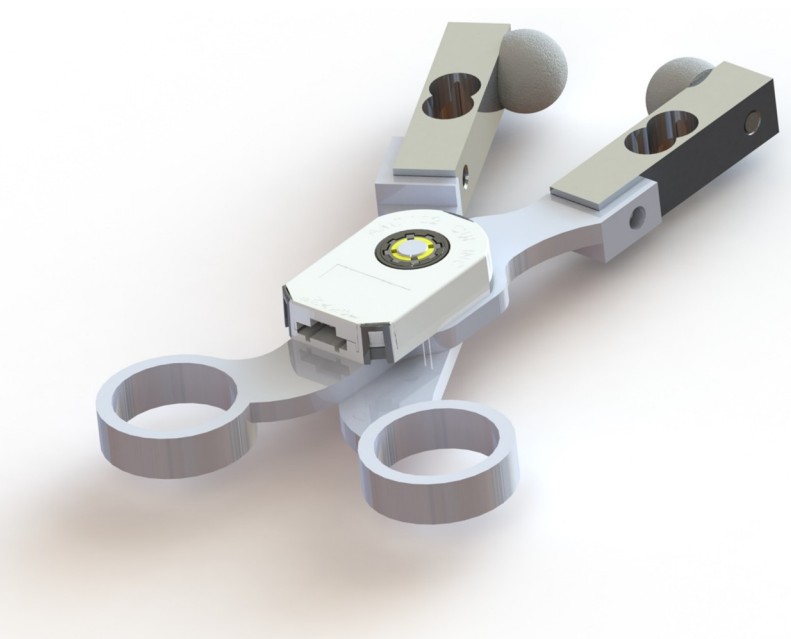

**Fig 2. Rendered image of the handheld grasper device.**

The grasping device was connected to a National Instruments DAQ, which was then connected to a computer. The computer ran the "Scissors Console" application, which sampled the voltage and encoder data at 1000 Hz and stored the data in CSV files defined by the user. Source code for the application can be found through S1 File. The application also displayed a real-time graph of the data, as shown in Drahos et al. [7]. We also used the application to calibrate the load cells, as detailed in S5 Fig.

**Usage.** Grasp data were collected from up to 20 different locations on the tissues at each testing state. Approximately half of the grasps were collected from previously grasped locations, with the exception of the *in vivo* state, as the tissues did not have any previous grasps during that testing state.

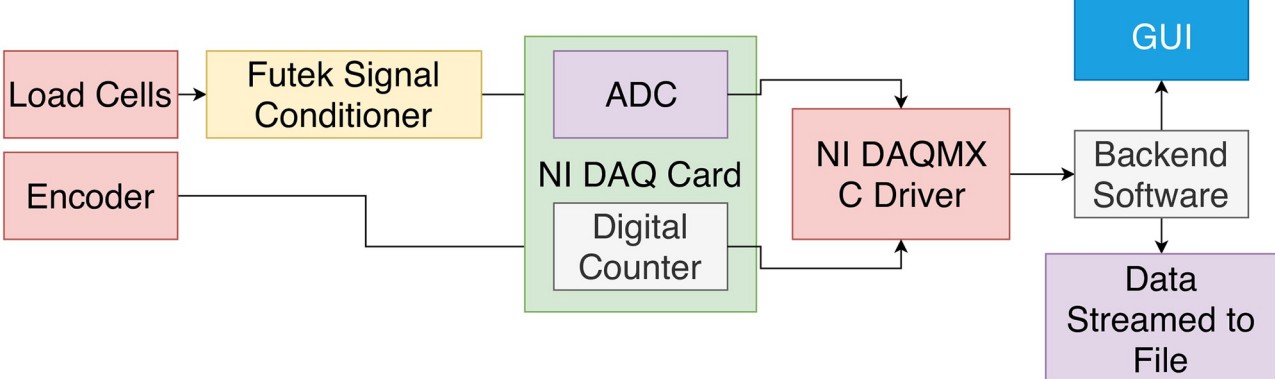

**Fig 3. The flow of data from the grasper sensors.**

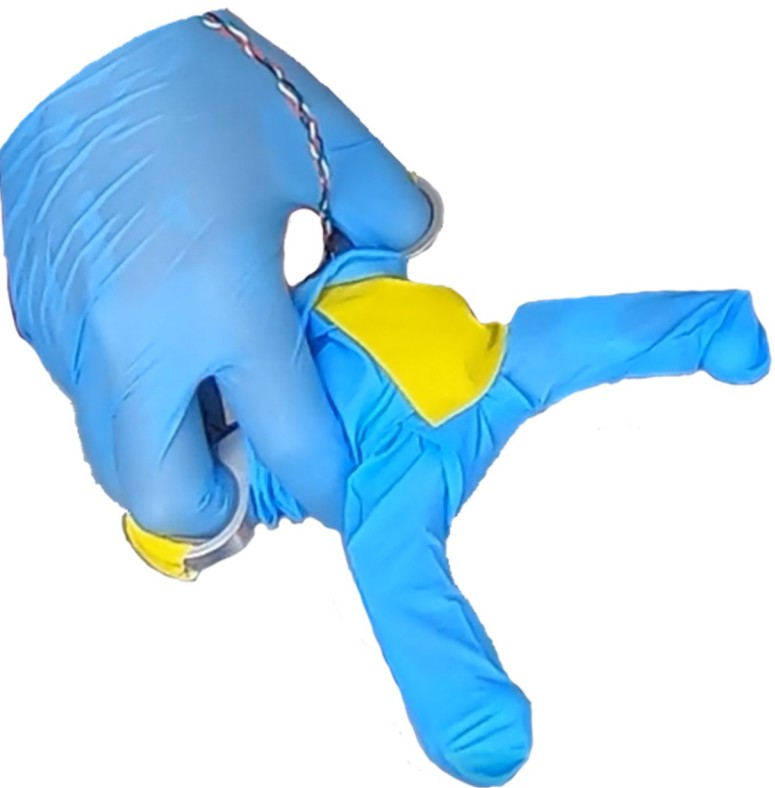

**Fig 4. A picture of the grasper device in gloves, being held by a gloved hand.**

Before collecting data from a specific location on the tissue, one individual (the "tissue holder"), positions and holds the tested tissue in order to facilitate grasping of the tissue at the specific location. During *in vivo* and *post mortem* tests, this often entailed holding parts of the tissue in more accessible regions of the abdominal cavity, in order to facilitate access to the tissue with the grasper. During the *ex vivo* and following tests, the tissue was simply held above the working area, keeping the tissue clear of obstructions and potential contaminants.

Before contacting the tissue with the grasper, the grasper tool was "stretched open" and closed three times in order to ensure the encoder index pulse is triggered. This also allowed us to verify that the grasp data was being successfully recorded by observing the "Scissors Console" data graph. After the "stretch" phase, the tissue was grasped 5–10 times at the specific location on the tissue, although only the first grasps at each location were used in this study. The "Scissors Console" application produced a series of audio tones at fixed time intervals to foster consistent grasp timings; a "grasp tone" was played for 1.5 seconds, with a delay of 1.7 seconds until the next tone.

After completing the series of grasps, data collection for the tissue test location was terminated, and the specific location was marked (using indelible black marker pens) to indicate that grasps were collected at the particular location. This study only uses the data from the first grasps collected from previously un-grasped tissue locations.

## Results

The force-displacement curves generated during grasping were plotted and utilized for curve-fitting, from which a stiffness value was derived for each grasp, as outlined in S4 Fig. Each

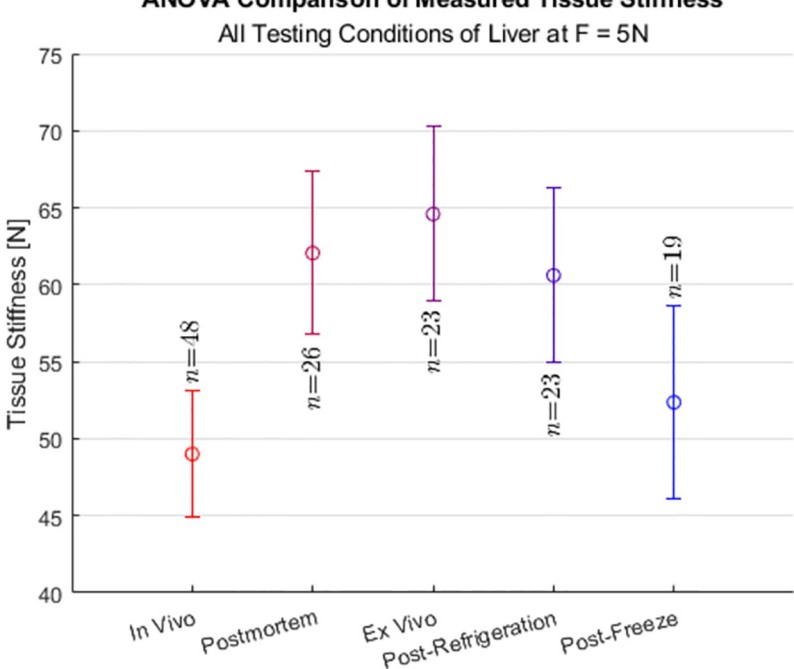

**Fig 5. ANOVA comparison of measured stiffness of the liver at F = 5N.** ANOVA comparisons ($p = 0.05$) for the Liver, comparing the *in vivo* tissue state to the other tissue states (*post mortem*, *ex vivo*, post-refrigeration, and post-freeze).

grasp was independently verified by two members of the research team to identify and exclude any invalid data, such as caused by equipment malfunctions or testing anomalies (e.g. tissue slippage, tissue puncture). The stiffness derivation methodology was validated using a calibration puck, as detailed in S6 Fig. Multiple one-way ANOVA comparisons ($p = 0.05$) were utilized to compare tissue stiffnesses from the *in vivo* tissue state to the other tissue states. We ensured that the tissue stiffnesses were normally distributed by utilizing the one-sample Kolmogorov-Smirnov test. All analysis was done using MATLAB. Grasps were excluded from stiffness analysis if they failed to meet the force threshold, or if curve-fitting quality (detailed in S4 Fig) was insufficient.

Fig 5 shows the results of the ANOVA comparisons for the Liver at a grasping force of 5 N. The changes in stiffness for the *post mortem*, *ex vivo*, and post-refrigeration states compared to the *in vivo* state were statistically significant. However, the change in stiffness from the *in vivo* to the post-freeze state was not statistically significant. The remaining discussion compares the *in vivo* and post-freeze states.

Fig 6(A) and 6(B) show that mean *in vivo* liver stiffnesses were fairly consistent across experimental animals; however, post-freeze stiffnesses varied substantially (generally increased stiffness). Mean peritoneal stiffnesses shown in Fig 6(C) and 6(D) varied substantially from animal to animal; however, the post-freeze stiffness for each animal was generally consistent with the *in vivo* data. The average change in stiffness from *in vivo* to post-freeze conditions for each tissue was relatively small compared to the measured tissue stiffnesses, as shown by the "average delta".

Fig 7 shows the one-way ANOVA confidence intervals ($p = 0.05$) for the measured tissue stiffnesses from *in vivo* to post-freeze for the Liver and Peritoneum. The results indicate that

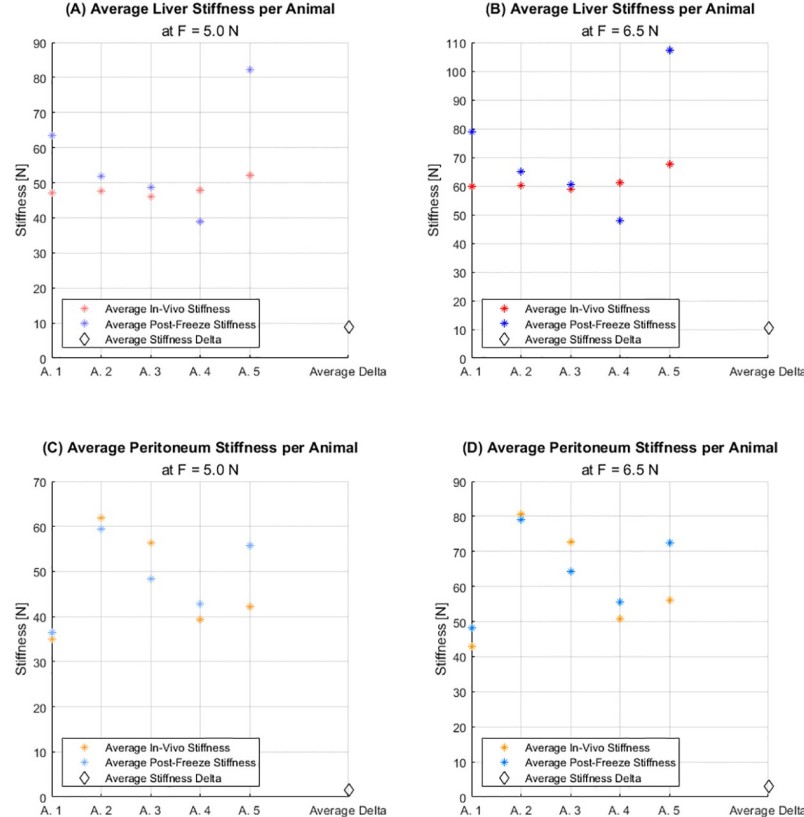

**Fig 6. Average measured tissue stiffnesses during *in vivo* and post-freeze, per animal.** (A) Liver at F = 5N. (B) Liver at F = 6.5N. (C) Peritoneum at F = 5N. (D) Peritoneum at F = 6.5N.

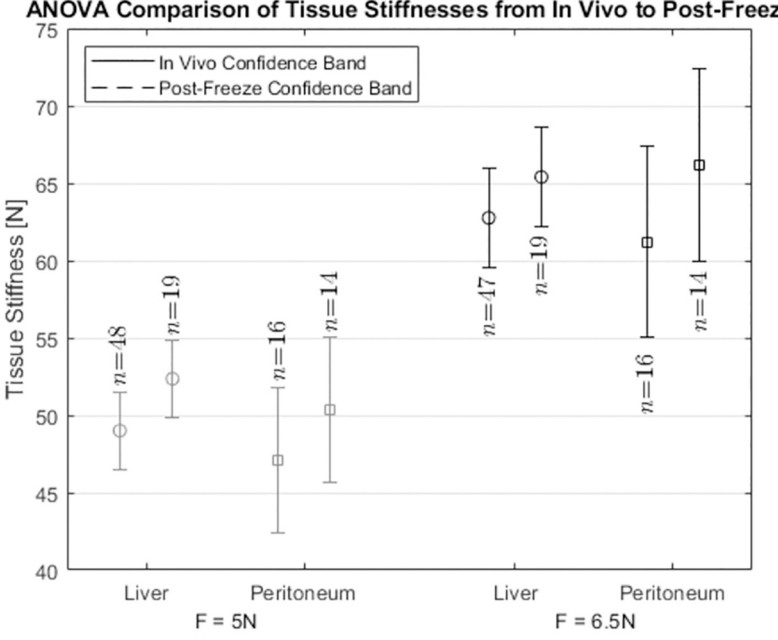

**Fig 7. ANOVA comparison of Liver and Peritoneum from *in vivo* to post-freeze, at both F = 5N and F = 6.5N.** Stiffnesses taken at lower force values are more faded (left), and higher force values are more saturated (right). The *in vivo* condition for each tissue is shown in solid lines, whereas the post-freeze condition is shown in dashed lines.

the changes in stiffness from *in vivo* to post-freeze conditions were not statistically significant. The variability in stiffness of the tissue under a particular condition (*in vivo* or post-freeze) was comparable with or exceeded the average change in stiffness from changing the tissue condition from *in vivo* to post-freeze.

## Discussion

The measured change in stiffness of Liver tissues is statistically significant under the *post mortem*, *ex vivo*, and post-refrigeration states compared to the *in vivo* state; however, the measured change in stiffness under the post-freeze (preserved) state is not statistically significant. This agrees with the observations made in Rosen et al. [1], Kerdok et al. [4], and Brouwer et al. [5], which all observed significant changes from *in vivo* to *ex vivo* tissues. This also agrees with Safdari et al. [6], which found that post-freeze liver and spleen tissue exhibited measurable, statistically significant lower stiffnesses compared to *post mortem* tissue.

Rosen et al. [1] found that stresses were higher at a set strain *in vivo*. While these results were a key finding, this study noted its limitation to phenomenological models and measurements (i.e., curve fitting), like this study. It did not reveal fundamental underlying mechanisms that caused the measured changes. However, such phenomenological measurements or models may suffice for medical simulators as a basic means to measure whether a simulated physical tissue matches its real living target in a quantifiable manner.

The primary conclusion from Kerdok et al. [4] is that there was a statistically significant difference between *in vivo* tissue and *ex vivo* post-perfused tissue. However, *in vivo* tissue had similar viscoelastic properties to *ex vivo* perfused tissue, with minimal differences being attributed to inconsistent perfusion pressures and concentrations.

Brouwer et al. [5] determined that a statistically significant discrepancy existed among fit parameters between *in vivo* and *ex vivo* tissue. This discrepancy was attributed to changes in boundary conditions from *in vivo* to *ex vivo*. These findings agree with the statistically significant changes in tissue stiffness in both liver and peritoneal tissue observed in this study.

Ocal et al. [3] found that as tissue was stored *post mortem*, the loss and relaxation moduli of the tissue increased. This indicates that as the tissue is stored over time the viscosity and stiffness of the tissue increases, which agrees with our results. However, our measured stiffnesses decreased, on average, going to post-refrigeration then again at post-freeze. This discrepancy is possibly explained by Kerdok et al. [4], which observed that perfused *ex vivo* tissue behaved similarly to *in vivo* tissue. We also observed that post-freeze stiffnesses were similar to *in vivo* stiffnesses. This indicates that the preservation process used in this study may affect liver and peritoneal stiffnesses, and cause them to revert back to *in vivo* levels, perhaps due to perfusion of the preserving fluid. Thus, if these observations can be replicated for different tissues, existing tissue data measured from preserved tissues may match the characteristics of the corresponding *in vivo* tissues.

Our results contrast with Mazza et al. [2], which was unable to determine any statistically significant difference in stiffness between *in vivo* and *ex vivo* cervical tissue from their aspiration experiment. However, the authors did cite that the lack of statistical significance could potentially be due to large variability in the data, thus further analysis could yield differing results. The major outcomes of this current study, as well as related outcomes of the previously discussed studies [1–6], are summarized below in Table 4.

## Conclusions

This study findings provide evidence in support the use of preserved tissue to model realistic, *in vivo* tissue mechanical responses for medical training, at least for situations deemed

**Table 4. Outcomes of prior work.**

| Reference | Study Outcomes |
|---|---|
| Rosen et al. [1] | *In vivo* tissue maintained higher normalized stress under compression. |
| Mazza et al. [2] | No statistically significant differences from *in vivo* to *ex vivo*, perhaps due to large data variability. |
| Ocal [3] | Stored *post mortem* tissue becomes stiffer and more viscous over time. |
| Kerdok et al. [4] | *Ex vivo* tissue was stiffer and more viscous than *in vivo* tissue; perfused *ex vivo* tissue behaved similar to *in vivo* tissue. |
| Brouwer et al. [5] | Significantly different curve-fit parameters between *in vivo* and *ex vivo* data. |
| Safdari et al. [6] | Post-freeze tissue had significantly lower stiffness than *post mortem*. |
| Current Study | *Post mortem* tissue had increased stiffness from *in vivo*, but post-freeze tissue stiffness was similar to *in vivo* tissue. |

adequately represented by porcine peritoneal and liver tissues. The study found no statistically significant differences between *in vivo* and post-freeze stiffnesses to argue the contrary position: that *in vivo* data collection should be prioritized over post-freeze approaches if the property of interest is stiffness. This is fortuitous, as preserved tissue tends to be easier to acquire and much easier to characterize for mechanical properties. These properties may be (and have been) used to create medical simulators, which are becoming increasingly common and relevant. Furthermore, the advent of "haptically-enhanced" virtual medical simulators (with force feedback) would benefit from well-characterized tissue mechanical responses collected from such sources.

However, there are multiple limitations to this study. First, this study only considers a small sample population ($n = 5$) of relatively young porcine models. The small sample population for this study was determined to be comparable to other exploratory works cited (2) (4). Further, it targets the mechanical response of some tissues that are in no way representative of the complexity of animal physiology and its methodology is relatively agnostic to tissue thickness and surface conditions. There are other factors that affect the perception and handling experience of tissues, such as organ size, which may be altered after death. Other factors such as presence or volume of bodily fluids may also impact the experience of interacting with tissue.

Future work should analyze the effects of different preservation techniques on tissues, and consider more comprehensive testing on other tissues and tissue states–particularly tissues relevant to medical mannequins and medical simulators such as perfusion. Additionally, a robotically-actuated grasper could be used, using strain rates derived from this work to capture typical human grasp motions, in order to increase testing consistency while maintaining realistic human operating strain rates. This will also allow greater analysis of the intermediate tissue states, which did exhibit changes in stiffness that were statistically significant.

Furthermore, *in vivo* human data collection is necessary in order to characterize the variability in tissue properties within the same patient, and across different patients and evaluate whether the findings from porcine models generalize to humans, sustaining the conclusion that post-freeze data collection on suitably-preserved tissues may be justified in place of *in vivo* measurements. If this work found evidence of large differences between *in vivo* and post-freeze states, it may have helped scientifically justify the added risk to conducting such research in humans. However, the lack of such evidence from this work does not help justify this increased risk.

## Supporting information

**S1 Fig. Measured temperatures of liver and peritoneum pre- and post-testing.**
(PDF)

**S2 Fig. Scissors console source code.**
(PDF)

**S3 Fig. Grasper device details.**
(PDF)

**S4 Fig. Curve fitting methodology.**
(PDF)

**S5 Fig. Load cell calibration.**
(PDF)

**S6 Fig. Methodology validation using calibration puck.**
(PDF)

**S1 File. Grasper test on calibration puck.**
(PDF)

## Acknowledgments

The authors thank Drs. Paul Iazzo, Tinen Isles, and the staff at the Visible Heart Lab (VHL) at the University of Minnesota for providing valuable input, IACUC support, brokering animal access, and running the animal experiments in their facilities. The authors also acknowledge the input of Dr. Victor H. Barocas for discussion regarding analysis methodology (especially comparison at force levels or use of spherical indenters) and Mathew Kubala and Rebecca Smith for assisting in data collection.

## Author Contributions

**Conceptualization:** Amer M. Safdari, Mark V. Mazzeo, Jack E. Norfleet, Robert M. Sweet, Timothy M. Kowalewski.

**Data curation:** Faizan A. Malik, Bradley A. Drahos, Amer M. Safdari.

**Formal analysis:** Faizan A. Malik, Bradley A. Drahos, Timothy M. Kowalewski.

**Funding acquisition:** Mark V. Mazzeo, Jack E. Norfleet, Robert M. Sweet, Timothy M. Kowalewski.

**Investigation:** Faizan A. Malik, Bradley A. Drahos, Amer M. Safdari, Mark V. Mazzeo, Jack E. Norfleet, Robert M. Sweet, Timothy M. Kowalewski.

**Methodology:** Bradley A. Drahos, Amer M. Safdari, Mark V. Mazzeo, Jack E. Norfleet, Robert M. Sweet, Timothy M. Kowalewski.

**Project administration:** Bradley A. Drahos, Mark V. Mazzeo, Jack E. Norfleet, Timothy M. Kowalewski.

**Resources:** Bradley A. Drahos, Amer M. Safdari, Robert M. Sweet, Timothy M. Kowalewski.

**Software:** Faizan A. Malik, Amer M. Safdari.

**Supervision:** Bradley A. Drahos, Mark V. Mazzeo, Jack E. Norfleet, Timothy M. Kowalewski.

**Validation:** Faizan A. Malik, Bradley A. Drahos, Timothy M. Kowalewski.

**Visualization:** Faizan A. Malik, Timothy M. Kowalewski.

**Writing – original draft:** Faizan A. Malik, Bradley A. Drahos, Timothy M. Kowalewski.

**Writing – review & editing:** Faizan A. Malik, Bradley A. Drahos, Amer M. Safdari, Mark V. Mazzeo, Jack E. Norfleet, Robert M. Sweet, Timothy M. Kowalewski.

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
