## [Decision Letter · Decision Letter 0]

27 Jul 2022

PONE-D-22-12980Variability of Tissue Mechanical Response in Sus Domesticus Porcine Models from in vivo to ex vivo ConditionsPLOS ONE

Dear Dr. Drahos,

Thank you for submitting your manuscript to PLOS ONE. After careful consideration, we feel that it has merit but does not fully meet PLOS ONE’s publication criteria as it currently stands. Therefore, we invite you to submit a revised version of the manuscript that addresses the points raised during the review process.I consider your article to be a valid contribution to this area of work, however I recommend some changes to the document, mostly language issues. The questions raised by the reviewers must be answered and their doubts clarified.

We look forward to receiving your revised manuscript.

Kind regards,

Carlos Alberto Antunes Viegas, DVM; MSc; PhD

Academic Editor

PLOS ONE

Journal Requirements:

2. As part of your revision, please complete and submit a copy of the Full ARRIVE 2.0 Guidelines checklist, a document that aims to improve experimental reporting and reproducibility of animal studies for purposes of post-publication data analysis and reproducibility: https://arriveguidelines.org/sites/arrive/files/Author%20Checklist%20-%20Full.pdf (PDF). Please include your completed checklist as a Supporting Information file. Note that if your paper is accepted for publication, this checklist will be published as part of your article

Additional Editor Comments (if provided):

Dear authors,

In order to improve the final version I recommend these changes:

Introduction

Pay attention to Latin terms such as in vivo which must be written in italics.

Post mortem in Latin should be written in italics. The authors use the Latin expression and the expressions postmortem or post-mortem that in my opinion should not be used.

Statistical p should also be written in italics.

Why do you write post-freeze in italics?

Table 2 is not previously announced in the text.

Material and Methods

Although Table 3 is a repository of the sample of animals used, the sample must be briefly described in advance.

The in vivo testing procedure is not described.

I am of the opinion that the description of the functioning of the test device should be made after the description of the sample and prior to the description of the studies.

Results

Line 295 – we are talking about experimental animals, not patients

Discussion

It is supposed in this section to compare our work with previous works.

Reviewers' comments:

Reviewer's Responses to Questions

**Comments to the Author**

1. Is the manuscript technically sound, and do the data support the conclusions?

Reviewer #1: Partly

Reviewer #2: Partly

2. Has the statistical analysis been performed appropriately and rigorously? 

Reviewer #1: No

Reviewer #2: Yes

3. Have the authors made all data underlying the findings in their manuscript fully available?

Reviewer #1: Yes

Reviewer #2: Yes

4. Is the manuscript presented in an intelligible fashion and written in standard English?

Reviewer #1: Yes

Reviewer #2: Yes

5. Review Comments to the Author

Reviewer #1: A number of issues exist with this manuscript which I will briefly point out.

1. There is no discussion

2. Introduction is a collection of paragraphs each describing one study

3. no statistical analysis is described within the graphs

4. table summarising previous studies (table 2) would benefit from more detail describing the outcome of those studies.

5. Table 3 would benefit to know the age of the animals (not haveing to calculate it one's self).

Reviewer #2: Dear author:

Congratulations for your effort in the preparation of this article. Reading the document I have some doubts that hope that you resolve me.

Lines 94-100. Rewrite the paragraph. You repeat sitiffness and viscosity twice in a short time. In my opinion not neccesary.

Line 104. Were the differences significant or not? I have not clear if it is mistoken.

Line 177. Please, chnage the commercial name of the drug for the correct (Tiletamine and zolazepam).

In addition, I have some doubts about the protocol:

1. I miss an anagesia protocol, specially in thoracic surgeries. The anesthesia is not enough and this may produce changes in the tissues caused by the shock.

2. You should explain what other procedures were done in these animals, because may have influence in your results.

3. How was calculated the sample size. If not, you should indicate it.

4. Why did you administered omega-3 fatty acids in some subjects and not in all. Did you check if this could have influence in the results.

6. PLOS authors have the option to publish the peer review history of their article (what does this mean?). If published, this will include your full peer review and any attached files.

Reviewer #1: No

Reviewer #2: No

---

## [Author Response · Author response to Decision Letter 0]

30 Sep 2022

For responses to all reviewer and editor comments please refer directly to the "Response to Reviewers.pdf" file. Each individual comment has been addressed including changes to the manuscript based on each piece of feedback.

---

## [Decision Letter · Decision Letter 1]

13 Jan 2023

PONE-D-22-12980R1Variability of Tissue Mechanical Response in Sus Domesticus Porcine Models from in vivo to ex vivo ConditionsPLOS ONE

Dear Dr. Drahos,

Thank you for submitting your manuscript to PLOS ONE. After careful consideration, we feel that it has merit but does not fully meet PLOS ONE’s publication criteria as it currently stands. Therefore, we invite you to submit a revised version of the manuscript that addresses the points raised during the review process.

We look forward to receiving your revised manuscript.

Kind regards,

Carlos Alberto Antunes Viegas, DVM; MSc; PhD

Academic Editor

PLOS ONE

Journal Requirements:

Reviewers' comments:

Reviewer's Responses to Questions

**Comments to the Author**

1. If the authors have adequately addressed your comments raised in a previous round of review and you feel that this manuscript is now acceptable for publication, you may indicate that here to bypass the “Comments to the Author” section, enter your conflict of interest statement in the “Confidential to Editor” section, and submit your "Accept" recommendation.

Reviewer #3: (No Response)

Reviewer #4: (No Response)

2. Is the manuscript technically sound, and do the data support the conclusions?

Reviewer #3: Yes

Reviewer #4: Partly

3. Has the statistical analysis been performed appropriately and rigorously? 

Reviewer #3: Yes

Reviewer #4: No

4. Have the authors made all data underlying the findings in their manuscript fully available?

Reviewer #3: Yes

Reviewer #4: Yes

5. Is the manuscript presented in an intelligible fashion and written in standard English?

Reviewer #3: Yes

Reviewer #4: Yes

6. Review Comments to the Author

Reviewer #3: The scientific paper "Variability of Tissue Mechanical Response in Sus Domesticus Porcine Models from in vivo to ex vivo Conditions" aimed to produce a comprehensive data set by attempting to measure the change in mechanical responses encountered in typical surgical grasping in porcine models between tissue in its native living (in vivo) state and four deceased states.

The authors made changes to the manuscript according to the reviewers' suggestions. In my opinion, the discussion is still the weak point of the manuscript, in addition to the scarce number of references used in this section.

Reviewer #4: The present paper is very important for healthcare simulators field as valuable resource for training technical and nontechnical skills.

I strongly recommend that authors amend the entire article to the form of a standard paper as Introduction, Material and Methods, Discussion and Conclusion.

1. Introduction with maximun 2 pages and containing the main objectives and justification;

2. I believe that the number of animals (n = 5) is quite small to have a reliable result with the main hypothesis of the study. If the authors want to keep this number, they will need to present the Test Power used in the methodology that indicated this minimum number of animals. It should be noted that sometimes the use of the minimum number of animals presented by the Power of Test is not ideal for the results to be faithful to the initial hypothesis of the study.

3. I strongly recommend that authors only perform tissue evaluation in vivo, ex vivo and post-freeze, as these are the most used methods in training; and in the following tissues: liver, spleen and lung.

7. PLOS authors have the option to publish the peer review history of their article (what does this mean?). If published, this will include your full peer review and any attached files.

Reviewer #3: No

Reviewer #4: No

---

## [Author Response · Author response to Decision Letter 1]

27 Feb 2023

Reviewer 3

The scientific paper "Variability of Tissue Mechanical Response in Sus Domesticus Porcine Models from in vivo to ex vivo Conditions" aimed to produce a comprehensive data set by attempting to measure the change in mechanical responses encountered in typical surgical grasping in porcine models between tissue in its native living (in vivo) state and four deceased states.

Thank you for taking the time to review our manuscript and provide your feedback. The research team has made various changes to the manuscript’s discussion based on your feedback which we feel strengthens the overall purpose and validity of the manuscript.

The authors made changes to the manuscript according to the reviewers' suggestions. In my opinion, the discussion is still the weak point of the manuscript, in addition to the scarce number of references used in this section.

Upon further review based on your feedback, as well as other reviewer’s feedback, the research team agrees that the discussion was the weakest section of the manuscript. The research team has reformatted the manuscript to better adhere to PLOS ONE standards and in doing so has restructured and added additional content to the discussion. Instead of presenting the results of the prior work in the introduction, the research team moved it to the discussion section to more directly compare with our results.

Reviewer 4

The present paper is very important for healthcare simulators field as valuable resource for training technical and nontechnical skills.

The research team agrees that the results from this study and manuscript bring attention to an important potential knowledge gap in regard to tissue properties. Thank you for taking the time to review our manuscript and provide your feedback. The team has made various edits to the overall structure of the manuscript based on your recommendations and have addressed some of your concerns.

I strongly recommend that authors amend the entire article to the form of a standard paper as Introduction, Material and Methods, Discussion and Conclusion.

The research team has restructured the manuscript to better adhere to PLOS ONE standards based on PLOS ONE guidelines and a review of other published manuscripts. The research team shortened the introduction. Instead of presenting the results of the prior work in the introduction, the research team moved it to the discussion section to more directly compare with our results.

1. Introduction with maximum 2 pages and containing the main objectives and justification;

While drafting the manuscript, the research team was not aware of this requirement and have adjusted the introduction to reduce its length to a more appropriate length. Thank you for bringing this to our attention.

2. I believe that the number of animals (n = 5) is quite small to have a reliable result with the main hypothesis of the study. If the authors want to keep this number, they will need to present the Test Power used in the methodology that indicated this minimum number of animals. It should be noted that sometimes the use of the minimum number of animals presented by the Power of Test is not ideal for the results to be faithful to the initial hypothesis of the study.

The research team acknowledges that the sample size used in this study is relatively small. A power test was conducted which determined between 15 and 30 animals would be required in order to achieve a statistical power of 0.8 to 0.95 across all tissues and tissue conditions. It is relatively common in animal-based exploratory studies to use a sample size lower than required for such statistical power. The research team has added verbiage to the conclusion section, in lines 419-421, of the manuscript stating this and refers to other cited work that makes use of similar sample sizes.

3. I strongly recommend that authors only perform tissue evaluation in vivo, ex vivo and post-freeze, as these are the most used methods in training; and in the following tissues: liver, spleen and lung.

The research team strove to be as comprehensive as possible with the resources available; the intermediate “post mortem'' and “post-refrigeration” states have been included in the manuscript as they may provide useful insight and comparisons for future work.

The research team did also gather data from spleen and lung tissues, however there was insufficient analyzable data for our analysis methods. This is due to difficulties in handling the tissues, largely caused by limited tissue surface area for grasping. Only previously un-grasped areas were used, as mentioned in lines 300-301.

---

## [Editor Report · Decision Letter 2]

1 Mar 2023

Variability of Tissue Mechanical Response in Sus Domesticus Porcine Models from in vivo to ex vivo Conditions

PONE-D-22-12980R2

Dear Dr. Bradley A. Drahos,

We’re pleased to inform you that your manuscript has been judged scientifically suitable for publication and will be formally accepted for publication once it meets all outstanding technical requirements.

Kind regards,

Carlos Alberto Antunes Viegas, DVM; MSc; PhD

Academic Editor

PLOS ONE
---

## [Editor Report · Acceptance letter]

6 Mar 2023

PONE-D-22-12980R2 

Variability of Tissue Mechanical Response in *Sus Domesticus* Porcine Models from *in vivo* to *ex vivo* Conditions 

Dear Dr. Drahos:

I'm pleased to inform you that your manuscript has been deemed suitable for publication in PLOS ONE. Congratulations! Your manuscript is now with our production department. 

Kind regards, 

on behalf of

Dr. Carlos Alberto Antunes Viegas 

Academic Editor

PLOS ONE